# Is Shiga Toxin-Producing *Escherichia coli* O45 No Longer a Food Safety Threat? The Danger is Still Out There

**DOI:** 10.3390/microorganisms8050782

**Published:** 2020-05-22

**Authors:** Yujie Zhang, Yen-Te Liao, Xiaohong Sun, Vivian C.H. Wu

**Affiliations:** 1College of Food Science & Technology, Shanghai Ocean University, Shanghai 201306, China; yujie.zhang@usda.gov (Y.Z.); xhsun@shou.edu.cn (X.S.); 2Produce Safety and Microbiology Research Unit, U.S. Department of Agriculture, Agricultural Research Service, Western Regional Research Center, Albany, CA 94710, USA; yen-te.liao@usda.gov

**Keywords:** Shiga-toxin producing *Escherichia coli* O45, comparative genomics, locus of enterocyte effacement, Stx prophages

## Abstract

Many Shiga toxin-producing *Escherichia coli* (STEC) strains, including the serogroups of O157 and most of the top six non-O157 serotypes, are frequently associated with foodborne outbreaks. Therefore, they have been extensively studied using next-generation sequencing technology. However, related information regarding STEC O45 strains is scarce. In this study, three environmental *E. coli* O45:H16 strains (RM11911, RM13745, and RM13752) and one clinical *E. coli* O45:H2 strain (SJ7) were sequenced and used to characterize virulence factors using two reference *E. coli* O45:H2 strains of clinical origin. Subsequently, whole-genome-based phylogenetic analysis was conducted for the six STEC O45 strains and nine other reference STEC genomes, in order to evaluate their evolutionary relationship. The results show that one locus of enterocyte effacement pathogenicity island was found in all three STEC O45:H2 strains, but not in the STEC O45:H16 strains. Additionally, *E. coli* O45:H2 strains were evolutionarily close to *E. coli* O103:H2 strains, sharing high homology in terms of virulence factors, such as Stx prophages, but were distinct from *E. coli* O45:H16 strains. The findings show that *E. coli* O45:H2 may be as virulent as *E. coli* O103:H2, which is frequently associated with severe illness and can provide genomic evidence to facilitate STEC surveillance.

## 1. Introduction

Shiga toxin-producing *Escherichia coli* (STEC), as one of the major foodborne pathogens, can produce Shiga toxins and cause severe human disease, such as diarrhea, hemorrhagic colitis, and hemolytic–uremic syndrome (HUS) [1,2,3]. STEC strains have been characterized by serotyping based on the O-antigen, determined by the polysaccharide portion of the cell wall lipopolysaccharide (LPS) and the H-antigen related to the flagella protein [4]. *E. coli* O157 and the “top six” non-O157 serogroups, including O26, O45, O103, O111, O121, and O145, are the primary STEC pathotypes frequently associated with foodborne outbreaks around the world [5,6]. Among the non-O157 STEC serotypes, STEC O45 has been identified as a cause of sporadic cases of bloody diarrhea [1]. In 2005, the first STEC O45 outbreak occurred in New York City, in which a total of 52 inmates were sick with diarrhea or bloody diarrhea, likely resulting from exposure to an ill food worker [7,8]. Subsequently, two *E. coli* O45:H2-associated outbreaks were reported to have caused 18 illnesses in the United States, and contaminated smoked game meat and goats were implicated as sources of contamination in these outbreaks [9].

The extensive research into STEC has utilized the advantages conferred by whole-genome sequencing technology and genomic analyses in the characterization of genetic information and the virulence factors associated with the diversity and evolution of STEC pathogens [10,11,12]. Bielaszewska et al. found that the *E. coli* O104:H4 strain associated with a large outbreak in Germany in 2011 was a clone of the *E. coli* O104:H4 (HUSEC041) strain isolated from the German outbreak in 2001 [13]. Another study reported that two *E. coli* O145:H28 outbreak strains (RM13714 and RM13716), linked to a U.S. lettuce-associated outbreak and a Belgium ice cream-associated outbreak, respectively, shared a common lineage with five different STEC O157:H7 strains, including the *E. coli* Sakai strain implicated in a Japanese outbreak [14,15]. Additionally, Ju et al. (2012) conducted phylogenetic analysis on bacterial whole-genome sequencing of STEC O26, O111, and O103 strains, and found a close phylogenetic relationship between *E. coli* O26:H11 and *E. coli* O111:H11 strains [16].

Other studies have demonstrated that the pathogenicity evolution of most STEC strains was highly associated with the acquisition of mobile virulence elements, such as Stx prophages, virulence plasmids, and pathogenetic genomic islands [13,14,17]. Specifically, the primary virulence factors of STEC pathogens, *stx* genes, were located on the Stx prophage sequences and could be transferred to other susceptible strains by phage transduction [18,19]. For example, two HUS-associated STEC O111:H^−^ strains (95JB1 and 95NR1) shared high genomic homology; however, the strain 95NR1 contained two additional Stx2 prophages that contributed to a higher HUS morbidity than 95JB1 [20]. In addition, the locus of enterocyte effacement (LEE) pathogenicity islands containing *espB, espD*, *espA, tir*, and *eae*, were virulence factors of STEC strains in charge of the production of proteins associated with intimate adherence to intestinal epithelial cells and the formation of attaching-and-effacing (A/E) lesions [19,21,22]. Several studies have indicated that LEE pathogenicity islands could be horizontally transferred between two different *E. coli* strains through specific insert loci, including *selC*, *pheU*, and *pheV* in the bacterial chromosome [23,24]. Moreover, several LEE-positive STEC strains, including those of the O26, O103, O111, O145, and O157 serogroups, have been frequently isolated from HUS patients [25]. Therefore, these findings reveal that the aforementioned virulence genes are essential for the pathogenicity evolution of STEC pathogens and are highly associated with the development of severe human illness, such as HUS [14,26].

Due to the close association with foodborne outbreaks and human illness, the U.S. Department of Agriculture (USDA) Food Safety and Inspection Service (FSIS) has declared *E. coli* O157 and the “top six” non-O157 serogroups of STEC, including O26, O45, O103, O111, O121, and O145, to have zero tolerance in raw ground beef and tenderized beef [27]. Many studies regarding the genomic characterization and evolutionary relatedness of these STEC strains have been conducted to minimize the spread of the pathogens and to enhance STEC surveillance related to foodborne outbreaks [28,29,30]. However, similar information about STEC O45 is lacking. Therefore, the objectives of this study were to characterize four STEC O45 strains isolated from environmental and clinical samples using next-generation sequencing technology, and to evaluate the pathogenicity of these STEC O45 strains in comparison to other clinical and outbreak-associated STEC strains.

## 2. Materials and Methods

### 2.1. Bacterial Strains

Three environmental *E. coli* O45:H16 strains (RM11911, RM13745, and RM13752) and one clinical *E. coli* O45:H2 strain (SJ7) were subject to whole-genome sequencing in the present study. The *E. coli* O45:H2 strain SJ7 was obtained from the Centers for Disease Control and Prevention (CDC), and was originally derived from a patient’s stool. The *E. coli* O45:H16 strain RM11911 was isolated from a standing water sample, and two *E. coli* O45:H16 strains (RM13745 and RM13752) were isolated from cattle feces in California. All strains were stored in the culture collections of the Produce Safety and Microbiology Research Unit in the USDA, Agricultural Research Service (ARS), Western Regional Research Center (WRRC). All strains were stored at −80 °C and grown in 10 mL of tryptic soy broth (TSB, Difco, Becton Dickinson, Sparks, MD) in a sterile 15-mL conical tube at 37 °C with shaking at 175 rpm before use.

### 2.2. Whole-Genome Sequencing, Assembly, and Annotation of STEC O45 Strains

Genomic DNA was extracted from the cultures grown to the mid-exponential-phase in 10 mL TSB using a Quick-DNA Miniprep Plus Kit (Zymo Research, Irvine, CA), according to the manufacturer’s instructions. The DNA library was constructed using a SMARTbell Express Template Prep kit 2.0 (PacBio, Menlo Park, CA) and subsequently sequenced on a Pacific Biosciences Sequel II with v1 reagents (PacBio, Menlo Park, CA) by SNPsaurus (Eugene, OR) [31]. An average of 251,060 single-end reads per bacterial strain were generated and subjected to de novo assembly using Flye 2.4.1 with default parameters. The resulting assemblies of each strain were generated with more than 100× coverage. The genome completeness of four strains was analyzed by BUSCO 3.0.2, and each assembled genome reached a score of above 98% completeness, which was identified as a complete genome [32]. The assembled contigs were also confirmed as bacterial chromosomes or plasmids by the use of BLASTn (https://blast.ncbi.nlm.nih.gov/Blast.cgi?PAGE=Nucleotides) [33]. The resulting genome sequences were subject to annotation using the National Center for Biotechnology Information (NCBI) Prokaryotic Genome Automatic Annotation Pipeline (PGAAP) (https://www.ncbi.nlm.nih.gov/genome/annotation_prok/) and deposited in GenBank. The circular maps were generated using the CGview server [34]. The accession numbers of the chromosome and plasmids for *E. coli* O45:H2 (SJ7) and three *E. coli* O45:H16 strains (RM11911, RM13745, and RM13752) are included in Table 1.

### 2.3. Determination of Virulence and Antibiotic Resistance Genes

Two reference *E. coli* O45:H2 strains (FWSEC0003 and 2011C-4251) were obtained from the NCBI database to facilitate the genomic analyses of virulence and antibiotic resistance genes of STEC O45 strains. *E. coli* clinical O45:H2 (FWSEC0003) was previously isolated from human feces in Canada [35], and the *E. coli* O45:H2 strain (2011C-4251) was isolated from samples of human feces obtained in the United States [36]. The virulence gene profiles of six strains (four experimental strains and two reference strains) were obtained using VirulenceFinder 2.0 [37]. The prediction of antibiotic resistance genes in each bacterial chromosome was conducted using ResFinder 3.2 [38] and Abricate 0.5 (https://github.com/tseemann/abricate). The setting with a minimum of both 95% nucleotide sequence identity and coverage was used to screen for these genes.

### 2.4. Identification of the Mobile Genetic Elements: Prophages and Genomic Islands

The identification of prophage and prophage-like sequences from each bacterial genome was conducted using the PHASTER web server [39]. Genomic islands were analyzed using the IslandViewer4 web server [40]. The default parameters were used for all the software.

### 2.5. Whole-Genome-Based Phylogenetic Analysis

Six STEC O45 strains, including four strains sequenced in this study and two reference strains from the NCBI database, as well as nine different STEC strains of clinical and outbreak origin (Appendix A) were subject to phylogenetic analysis by whole-genome multilocus typing (wg-MLST). Specifically, a total of 14,837 loci and 2513 core loci located on the entire genomes of 15 *E. coli* strains were analyzed and phylogenetic networks were constructed via a complete linkage method using BioNumerics 7.6 (Applied Maths, Kortrijk, Belgium). The GenBank accession numbers of 11 reference bacterial genomes obtained from the public database are indicated in Appendix A.

### 2.6. Comparative Genomics of E. coli O45:H2 and E. coli O103:H2

Based on the results of wg-MLST, two reference *E. coli* O103:H2 strains (12009 and 2015C-3163) of clinical origin were subject to comparative genomics analysis with STEC O45:H2 on the virulence genes and mobile genetic elements to evaluate their genetic relatedness using the BLAST Ring Image Generator (BRIG) with a minimum nucleotide sequence identity of 50% [41]. The crucial virulence factors of STEC pathogens, including Stx prophages and LEE pathogenicity islands, were extracted from each genome and subsequently aligned using the MAFFT algorithm in Geneious (Version 11.1.5, Biomatters, New Zealand). The phylogenetic trees of Stx prophages were constructed using Mega-X (Version 10.0.5) with the maximum likelihood algorithm and visualized using the Interactive Tree Of Life (ITOL) webserver [42]. The aligned sequences and the consensus identity of the LEE pathogenicity islands were visualized using Geneious (Version 11.1.5, Biomatters, New Zealand) [43,44].

## 3. Results

### 3.1. Genomic Features of Three E. coli O45:H16 Environmental Strains

The chromosomes of the three *E. coli* O45:H16 strains RM11911, RM13745, and RM13752 had genome sizes of 5,310,338, 5,264,698, and 5,264,517 bp, containing a total of 5194, 5264, and 5264 regions of coding DNA sequence (CDS), respectively; 91% of which were annotated with known functions (Table 1). RM11911, RM13745, and RM13752 strains also contained 22 rRNA and 90–91 tRNAs in each chromosome. Regarding the prediction of numerous mobile elements, RM11911 harbored 14 prophages, including a 65,626 bp Stx prophage carrying an *stx*_1a_ gene, whereas both RM13745 and RM13752 contained 12 prophages, including a 65,626 bp Stx1a prophage located at the same insertion site of each bacterial genome. Three *E. coli* O45:H16 strains also contained two additional virulence factors, *iroN* and *lpfA* genes, which were associated with the utilization of the siderophore enterobactin and bacterial adhesion and colonization in the intestine, respectively. Additionally, the antibiotic resistance gene *mdfA* was detected in each chromosome of *E. coli* O45:H16, which was attributed to a broad spectrum of drug-resistant features of these strains. 

The RM11911 strain contained one large circular plasmid, pRM11911, which was 175,089 bp in length, with an average G+C content of 47.6%. The plasmid pRM11911 also harbored several virulence factors, including the *hyl* operon (*hylC, hylA, hylB*, and *hylD*) encoding hemolysin, *afa* family (*afaA-VIII, afaB-VIII, afaC-VIII, afaD-VIII, afaE-VIII*, and *afaF-VII*) encoding adhesins, *cdt* family (*cdtA*, *cdtB*, and *cdtC*) encoding cytolethal distending toxin B, *espP* encoding extracellular serine protease, *cnf1* encoding cytotoxic necrotizing factor 1, and *iha* encoding the IrgA homolog adhesin (Table 2). The RM13745 strain harbors two plasmids: pRM13745-1 (45,063 bp) and pRM13745-2 (70,650 bp). The virulence gene-screening results showed that *espP* and *afa* families were present in pRM13745-1, while *iha* and *hyl* operons were present in pRM13745-2. RM13752 had one plasmid (pRM13752), with a genome size of 99,009 bp that contained the virulence genes, including *hyl* operon and *cnf1*. No antibiotic resistance genes were found in any of these plasmids.

### 3.2. Genomic Features of a Clinical E. coli O45:H2 Strain

The *E. coli* O45:H2 strain SJ7, isolated from a patient stool, contained a 5,444,105 bp chromosome which was approximately 130–180 kb smaller than the genomes of the three *E. coli* O45:H16 strains (Table 1). The chromosome of SJ7 consisted of 5278 CDSs, 22 rRNA, and 96 tRNA (Figure 1a). Ninety-one percent of CDSs were annotated with known functions. A total of 17 prophages, including one Stx1a prophage which was 65,813 bp in length, were predicted in the bacterial chromosome. Most of all, strain SJ7 contained a LEE pathogenicity island integrated at the *pheV* tRNA locus. Several non-LEE-encoded type III translocated virulence genes, including *nelA*, *nelB*, *nelC*, *espI*, and *cif*, were also present in the chromosome. No antibiotic resistance genes were found in the genome of SJ7.

SJ7 had two plasmids: pSJ7-1 and pSJ7-2 (Figure 1b,1c, Table 2). Plasmid pSJ7-1 had a genome of 74,390 bp with an average GC content of 49.1% and three virulence genes, including *stcE* (encoding metalloprotease), *ehxA* (encoding enterhemolysin), and *etpD* (encoding type II secretion protein). The second-largest plasmid, pSJ7-2, was 53,543 bp in length with 49.2% GC content, and contained no virulence genes. Neither plasmids harbored antibiotic resistance genes.

The two other clinical *E. coli* O45:H2 strains (FWSEC0003 and 2011C-4251) were obtained from the NCBI database to facilitate the genomic characterization of STEC O45:H2 strains (Table 1). Similar to the strain SJ7, FWSEC0003 and 2011C-4251 had a genome size of 5,532,455 and 5,440,026 bp, respectively, with an average GC content of 50.7% and a similar number of CDSs (>5700). FWSEC0003 contained an Stx1a prophage (56,560 bp), whereas 2011C-4251 had two Stx prophages consisting of a 64,500 bp Stx1a prophage and a 62,499 bp Stx2a prophage. Furthermore, both FWSEC0003 and 2011C-4251 strains encoded the LEE pathogenicity island, and several non-LEE-encoded type III translocated virulence factors, including *nelA*, *nelB*, *espI*, and *cif* (Table 2).

2011C-4251 had a plasmid (68,062 bp) which contained the virulence genes *ehxA, etpD*, and *stcE*, similar to SJ7. FWSEC0003 had two plasmids (95,228 and 52,940 bp), and the virulence genes *etpD* and *stcE* were found in the smaller plasmid. Furthermore, no antibiotic resistance genes were found in the plasmids of these two strains (Table 2).

### 3.3. Phylogenetic Analysis of 15 STEC Strains Using Whole-Genome Multilocus Typing

The evolutionary relatedness of 15 STEC strains, including four *E. coli* O45 strains (RM11911, RM13745, RM13752, and SJ7) sequenced in this study, two reference *E. coli* O45 strains (FWSEC0003 and 2011C-4251), and nine reference STEC strains with different serotypes, was investigated through whole-genome multilocus typing analysis. The results show that these STEC strains grouped into five clusters in the phylogenetic tree (Figure 2). Two outbreak strains, *E. coli* O145:H28 (RM13514) and *E. coli* O157:H7 (Sakai), grouped into cluster 1, while a clinical *E. coli* O103:H25 (2013C-3264) and an outbreak-associated *E. coli* O121:H19 (16-9255) strain were located in cluster 2. Additionally, cluster 3 contained a clinical *E. coli* O103:H11 strain (2013C-4225) and two outbreak strains of *E. coli* O26:H11 (11368) and *E. coli* O104:H4 (2011C-3493). Cluster 4 was composed of three *E. coli* O45:H2 strains (SJ7, FWSEC0003, and 2011C-4251) and two clinical *E. coli* O103:H2 strains (12009 and 2015C-3163), whereas three environmental *E. coli* O45:H16 strains (RM11911, RM13745, and RM13752) grouped into cluster 5. Additionally, the results showed that the three clinical *E. coli* O45:H2 strains shared a common evolutionary lineage with two *E. coli* O103:H2 strains (cluster 4), but differed from the environmental *E. coli* O45:H16 strain (cluster 5) and other STEC strains of clinical and outbreak origin (Figure 3). The findings indicate significant genomic differences between the *E. coli* O45:H16 and *E. coli* O45:H2 strains.

### 3.4. Comparative Genomics of E. coli O45:H2 and E. coli O103:H2 Strains

Due to the close genetic relatedness, the comparative genomics analysis was further performed on the backbones and mobile genetic elements of the three *E. coli* O45:H2 and two *E. coli* O103:H2 strains to investigate their evolutionary relationship. The whole-genome sequences of two *E. coli* O45:H2 (FWSEC0003 and 2011C-4251) reference strains and two *E. coli* O103:H2 (12009 and 2015C-3163) reference strains were compared against the *E. coli* O45:H2 strain (SJ7) using BLASTn, and the results show that the genomic backbones of these strains shared more than 95% identity covering over 95% of the SJ7 chromosome (Figure 3). Furthermore, non-homologous regions were identified among the genomes of *E. coli* O45:H2 and *E. coli* O103:H2 strains, particularly in the sequences of their mobile genetic elements. In the three *E. coli* O45:H2 chromosomes, prophage sequences were the primary non-homologous regions. For example, a 42,590-bp prophage (prophage_17) present in the genome of SJ7 was absent in both genomes of FWSEC0003 and 2011C-4251. Additionally, both FWSEC0003 and 2011C-4251 contained a prophage sequence sharing a low nucleotide sequence similarity to prophage_7 located in similar regions in the chromosome of SJ7. Another prophage sequence (prophage_6) in SJ7 shared low homology to the counterpart prophage sequence in 2011C-4251. On the other hand, genomic heterology of the mobile genetic elements was also observed for the *E. coli* O45:H2 strain (SJ7) and two *E. coli.* O103:H2 (12009 and 2015C-3163) strains. The prophage_1 and prophage_17 sequences found in SJ7 were absent in similar regions in the chromosomes of 2015C-3163 and 12009, respectively. Additionally, there was a region of low homology located at the sequence of genomic island_7 in SJ7 compared to the counterpart in both *E. coli* O103:H2 strains. Minor variations in specific regions, such as genomic island_1, prophage_7, and prophage_10, in *E. coli* O45:H2 (SJ7) were observed in comparison to both *E. coli* O103:H2 strains. These results indicate that the presence of different mobile genetic elements, and prophages in particular, was primarily attributed to the heterology of the closely related *E. coli* O45:H2 and *E. coli* O103:H2 strains.

### 3.5. Virulence Factors Located on Mobile Genetic Elements

The virulence factors of *E. coli* O45:H2 and *E. coli* O103:H2 were analyzed, and the results show that all virulence genes of each strain, except for *gad*, were located on different prophages and genomic islands (Figure 3). Stx prophages and LEE pathogenicity islands were among the most important virulence factors contributing to the pathogenicity of these pathogenic strains. The Stx prophages in the chromosomes of *E. coli* O45:H2 and *E. coli* O103:H2 were predicted and compared to investigate their genetic similarity (Figure 4). The phylogenetic analysis of Stx prophages showed that the Stx2a prophage in *E. coli* O103:H2 (12009) was classified as being alone in cluster 1. The Stx1a prophages from two *E. coli* O45:H2 strains (SJ7 and 2011C-4251) and two *E. coli* O103:H2 strains (12009 and 2015C-3163) were grouped into cluster 2. The Stx1a prophage and Stx2a prophage from *E. coli* O45:H2 (FWSEC0003) and *E. coli* O45:H2 (2011C-4251), respectively, were classified into cluster 3. Based on the phylogenetic results, the high sequence homology between the Stx1a prophage of *E. coli* O45:H2 strain (SJ7) and the two *E. coli* O103:H2 strains likely indicates that *E. coli* O45:H2 might exhibit a similar Stx prophage-associated pathogenicity to *E. coli* O103:H2.

Additionally, one of the essential virulence factors of STEC strains, the LEE pathogenicity island, was also present among the genomes of these *E. coli* O45:H2 and *E. coli* O103:H2 strains (Figure 5). LEE pathogenicity islands had an average sequence size ranging from 61,863 to 88,869 bp and were located at the *pheU* or *pheV* tRNA locus. All LEE pathogenicity islands contained the core virulence genes (*espF, espB, espA, eae*, and *tir*) in the highly conserved region (Figure 5). All strains, except for *E. coli* O103:H2 (2015C-3163), contained the LEE pathogenicity islands harboring extra virulence genes, *nleB* and *efa1*, which were located in the non-homologous region of the LEE pathogenicity island. The comparative genomics of LEE pathogenicity islands from the selected strains also showed that the *E. coli* O103:H2 strain (12009), associated with a diarrhea patient from a previous foodborne infection in Japan, and the three clinical *E. coli* O45:H2 strains (SJ7, FWSEC00033, and 2011C-4251) shared a high nucleotide sequence identity, demonstrating that these *E. coli* O45:H2 strains could harbor similar human pathogenesis, causing diarrhea.

## 4. Discussion

Due to the advancement of next-generation sequencing technology, the high accuracy of complete bacterial genome information has dramatically facilitated the genomic characterization of STEC pathogens and investigation of the evolutionary relationship between different pathogenic *E. coli* strains. A number of studies have utilized whole-genome sequencing technology to predict the parallel evolution of distinct STEC strains, including the O26, O111, O103, O145, and O157 serogroups [14,45]. Their findings have not only provided genomic evidence related to the clinical impact of these strains, but also facilitated the surveillance of these STEC serotypes. However, similar studies, including on published complete genomes of STEC O45 strains, are lacking. Currently, only two complete genome sequences of STEC O45:H2 strains (FWSEC0003 and 2011C-4251) are available in the public database. Therefore, in this study, four STEC O45 strains, from environmental and clinical samples, were sequenced and subjected to comprehensive genomic characterization, particularly for crucial virulence factors, and a comparison with other reference bacterial genomes was conducted to understand whether or not STEC O45 poses a similar food safety threat as the other serogroups.

### 4.1. STEC O45:H16 Strains Are Genomically Conserved

In this study, the results showed that the genome features of three *E. coli* O45:H16 strains (RM11911, RM13745, and RM13752) contained similar virulence factors, most of which were located on similar prophage sequences. All three *E. coli* O45:H16 strains contained the same Stx1a prophage (65,626 bp) with 100% nucleotide sequence homology (Figure 4). A previous study in our lab also demonstrated that Stx1a prophages from STEC O45 genomes were more conservative than those detected from other serotypes of STEC genomes [46]. Additionally, these three STEC O45:H16 strains contained a virulence factor—the *iss* gene—contributing to the serum resistance of *E. coli* strains [47,48]. A previous study revealed that the tested *E. coli* strains contained three *iss* alleles, which were likely disseminated among *E. coli* bacteria through horizontal gene transfer [49]. In this study, each of the three O45:H16 strains contained two different *iss* alleles, which were located on the Stx1a prophage and a suspected 50,703-bp prophage sequence, respectively. These findings show that the three environmental *E. coli* O45:H16 strains in this study were genomically conserved, with the virulence factors being carried by certain prophages.

### 4.2. STEC O45:H2 Evolved from a Common Ancestor with STEC O103:H2

The current results of wg-MLST analysis showed that three STEC O45:H2 strains (SJ7, FWSEC0003, 2011C-4251) and two STEC O103:H2 strains (12009 and 2015C-3163) have a close evolutionary relationship (Figure 2). Moreover, the comparative genomics of *E. coli* O45:H2 and *E. coli* O103:H2 revealed that the genomic backbone of these strains share 99% homology. Previous studies have demonstrated that a large percentage of *E. coli* O103:H2 strains have a high level of virulence in human infection and are commonly associated with diarrhea and HUS [50,51]. In particular, *E. coli* O45:H2 strains (SJ7, FWSEC0003, 2011C-4251) were shown to contain the crucial virulence factors, *eae* and *tir*, that are found in most STEC O103:H2 strains and are involved in the formation of A/E lesions [52]. Additionally, *E. coli* O45:H2 and *E. coli* O103:H2 used in this study were found to contain non-LEE-encoded type III translocated virulence genes, which are related to bacterial colonization and the development of HUS [53,54].

Additionally, the wg-MLST result in this study showed that *E. coli* O145:H28 (RM13514) and *E. coli* O157:H7 (Sakai) shared a close phylogenetic relationship, which is in agreement with the findings of a previous study demonstrating that STEC O145:H28 and O157:H7 strains evolved from a common evolutionary lineage [55]. The author also found that these two STEC serotypes could be further classified into sublineages based on the presence of different virulence factors, such as Stx prophages and the large virulence plasmids. However, in this study, the most highly heterogeneous regions between *E. coli* O45:H2 and *E. coli* O103:H2 strains fell on the sequences corresponding to prophages and genomic islands which were not associated with virulence factors (Figure 3). These findings likely suggest that due to a high homology of the virulence factors, *E. coli* O45:H2 may possess a similar pathogenicity to *E. coli* O103:H2 and could potentially cause severe human diseases such as HUS. Thus, the critical findings of the close genetic relatedness between the clinical STEC O103:H2 and STEC O45:H2 strains could facilitate future development of the antimicrobial strategies for the control of STEC O45 strains because some antimicrobial agents, such as weak acids, or lytic bacteriophages, effective in mitigating STEC O103 could also be tested to control STEC O45 strains [56,57]. Other natural antimicrobials that are capable of displaying a wide spectrum of antimicrobial activity and overcoming the emergence of antibiotic resistance in bacterial pathogens could also be considered [58].

### 4.3. Mobile Elements Play a Key Role in Driving the Virulence Evolution of STEC

Mobile genetic elements play a key role in shaping the bacterial genome and virulence evolution of STEC [59,60,61]. The current results also show that most virulence factors were located on mobile genetic elements, including plasmids, prophages, and genomic islands. In this study, the four STEC O45 strains contained different plasmids. Even the three *E. coli* O45:H16 environmental strains, which shared high nucleotide sequence similarity of the bacterial chromosomes, had plasmids with different genome sizes and contained various virulence genes, including *afa*, *cdt*, *esp*, *cnf1*, and *iha*. Several studies have indicated that the plasmids of LEE-negative STEC strains, which were found to contain encoded virulence factors, can also cause severe human diseases such as HUS, an ability which is frequently observed for LEE-positive STEC pathogens [62,63,64,65]. Additionally, a previous study conducted by Michelacci et al. reported that one LEE-negative STEC strain, isolated from an HUS patient, contained a plasmid harboring several virulence factors, including *ehxA*, *sta1*, and a novel variant of the *faeG*, in particular [64]. Notably, the *faeG*, encoding the production of ETEC F4 fimbriae, is commonly found in the pathogenic *E. coli* strains related to causing swine diseases, but not in those that could affect human health. Therefore, the presence of *faeG* in the plasmid of this human clinical strain provided genetic evidence for virulence gene transfer among plasmids of different STEC strains, which contributed to the pathogenicity evolution of the STEC strains.

Prophages represent an essential member of mobile genetic elements and are highly distributed in the genomes of different *E. coli* strains [66]. Many studies have indicated that *stx*, encoded in Stx prophage sequence, is the main virulence factor of STEC [46,67,68,69]. The results of this study showed that three *E. coli* O45:H16 strains carried a similar Stx prophage, whereas the phylogenetically related *E. coli* O45:H2 and *E. coli* O103:H2 strains carried distinct Stx prophages. The current results were supported by the finding of our previous study that the distribution of different Stx prophages is highly associated with specific STEC serotypes. Some serotypes of STEC strains, such as O103, were able to accept a wide range of different Stx prophages, but other serotypes, like O121, were only susceptible to infection with genetically conserved Stx prophages [46]. Therefore, the diversity of Stx prophages from *E. coli* O45:H2 (SJ7) and *E. coli* O103:H2 strains used in this study likely implies that these strains are susceptible to accepting exogenous genes, rather than *E. coli* O45:H16 strains. Additionally, other prophages predicted from all of the STEC O45 and O103 strains in this study also contained several virulence factors, including non-LEE encoded type III translocated effectors (*nleA, nleB, nleC*, and *cif*), serine protease autotransporters (*espI*), and increased serum survival gene (*iss*). Similar results were also found in a previous study, in which the genes encoding type III secretion in *E. coli* were located on a vast prophage region, which acted as a crucible for the evolution of pathogenicity in numerous *E. coli* species [70]. Interestingly, the results of this study show that the prophages carrying non-LEE encoded type III translocated effectors were only identified in LEE-positive strains, including *E. coli* O45:H2 and *E. coli* O103:H2 strains, but were absent in the *E. coli* O45:H16 LEE-negative strains. This phenomenon will need to be investigated in future studies.

## 5. Conclusions

This is the first study to report the genomic characterization of STEC O45 strains of different origins. The whole-genome-based phylogenetic analysis revealed that the environmental *E. coli* O45:H16 strains were phylogenetically distinct from the clinical *E. coli* O45:H2 strains, whereas both clinical *E. coli* O45:H2 and *E. coli* O103:H2 shared a common evolutionary ancestor. Furthermore, most of the crucial virulence factors from *E. coli* O45:H2 and *E. coli* O103:H2 shared high nucleotide sequence similarity and were located on mobile genetic elements, such as prophages, genomic islands, or plasmids, which is strongly associated with the pathogenicity evolution of these STEC strains. The findings of this study provide better insights into the genomic characterization, evolutionary relatedness, and virulence evolution of STEC O45 strains and indicate that *E. coli* O45:H2 strains still pose potential threats to public health and should, therefore, be included in epidemiological surveillance.

## Figures and Tables

**Figure 1 microorganisms-08-00782-f001:**
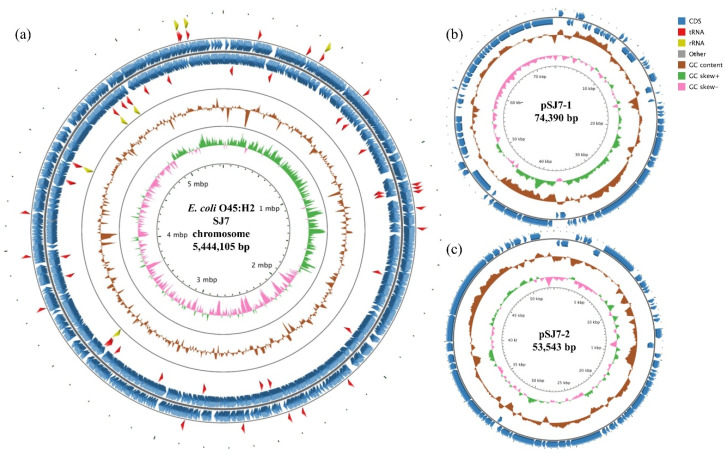
Circular genome maps of the chromosome (**a**), plasmid pSJ7-1 (**b**), and plasmid pSJ7-2 (**c**) for the clinical STEC O45:H2 strain SJ7 generated using CGview server. For the map of chromosome (a), the rings from the inside out represent the GC skew (green and pink), GC content (brown), coding DNA sequences (CDSs) (blue), tRNA (red), and rRNA (yellow). For the maps of two plasmids (pSJ7-1 and pSJ7-2), the same color codes are used, and no tRNA and rRNA were detected. The annotated functions of the CDSs are not indicated.

**Figure 2 microorganisms-08-00782-f002:**
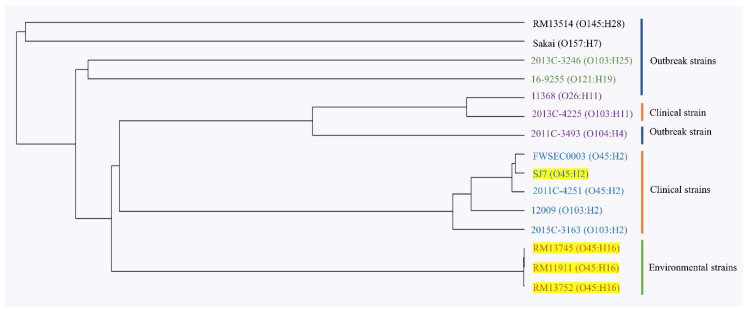
Whole-genome multilocus typing-based phylogenetic analysis of three environmental *E. coli* O45:H16 strains (RM13745, RM11911, and RM13752) and one clinical *E. coli* O45:H2 strain (SJ7) with eleven reference strains. The phylogenetic tree indicates five distinct clusters, with the strain names shown in black color for cluster 1, green color for cluster 2, purple color for cluster 3, blue cluster for cluster 4, and orange color for cluster 5. The strains highlighted in yellow were sequenced this study. The sources of isolation (outbreak, clinical, and environmental) for these 15 strains are indicated right next to the strain names.

**Figure 3 microorganisms-08-00782-f003:**
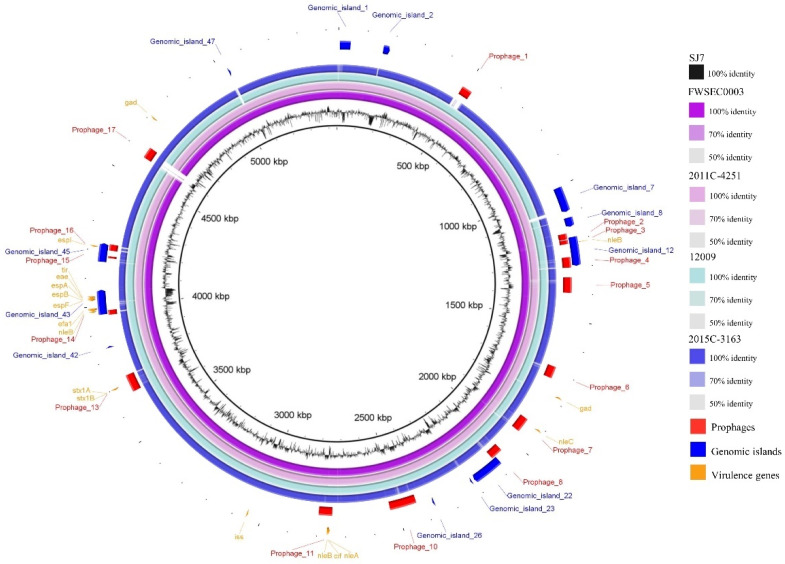
Comparative analysis of the complete chromosome of the STEC O45:H2 strain SJ7 with two STEC O45:H2 and two STEC O103:H2 reference strains. The BLAST comparison of STEC chromosomes and the chromosome of *E. coli* O45:H2 strain SJ7 was generated using the BLAST Ring Image Generator (BRIG) with a minimum nucleotide sequence identity of 50%. The bacterial chromosomes from inside out represent *E. coli* O45:H2 SJ7 (black), *E. coli* O45:H2 FWEC0003 (dark purple), *E. coli* O45:H2 2011C-4251 (light purple), *E. coli* O103:H2 12009 (light blue), and *E. coli* O103:H2 2015C-3163 (dark blue). Color codes for different degrees of nucleotide sequence homologies based on comparison to SJ7 are listed. The prophages (*red*), genomic islands (*navy*), and virulence genes (*yellow*) annotated in the SJ7 chromosome are present at the outermost three rings.

**Figure 4 microorganisms-08-00782-f004:**
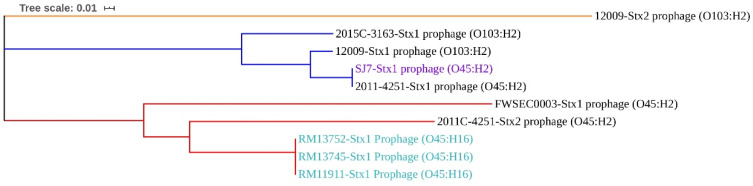
Maximum likelihood phylogenetic analysis based on the MAFFT algorithm of the Stx prophages from the genomes of six STEC O45 strains and two STEC O103:H2 strains. The phylogenetic tree indicates three distinct clusters among the Stx prophages, in orange for cluster 1, navy for cluster 2, and red for cluster 3. Stx prophages highlighted in purple and light blue indicates they were obtained from the bacterial strains sequenced in this study. All Stx prophage sequences in the bacterial chromosome were predicted using the PHASTER web server.

**Figure 5 microorganisms-08-00782-f005:**
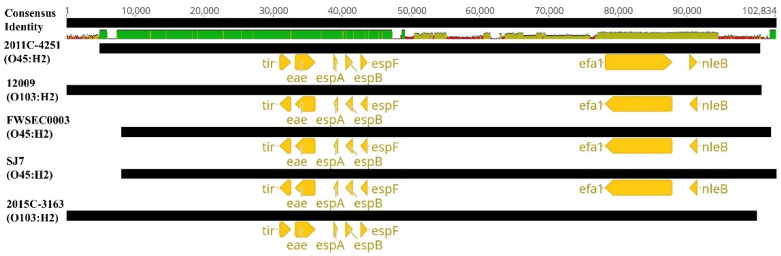
Nucleotide sequence alignments of the locus of enterocyte effacement (LEE) pathogenicity islands from the genomes of three clinical STEC O45:H2 strains (SJ7 (sequenced in this study), FWSEC003, and 2011C-4251), and two clinical STEC O103:H2 strains (12009 and 2015C-3163) using the MAFFT algorithm. The consensus identity of the alignment of five LEE pathogenicity islands indicates the mean pairwise nucleotide sequence identified for all pairs in the column: green = 100% identity; green-brown = <100% but >30% identity; and red = <30% identity. Virulence genes within the sequence of each LEE pathogenicity island *(black*) are annotated and colored (*yellow*).

**Table 1 microorganisms-08-00782-t001:** Genomic features of the chromosomes and plasmids for six Shiga toxin-producing *Escherichia coli* (STEC) O45 strains analyzed in this study, including three environmental and three clinical strains.

Feature	*E. coli* O45:H16			*E. coli* O45:H2		
	RM11911	RM13745 ^α^	RM13752	SJ7 ^α^	FWSEC0003 ^α^	2011C-4251
**Chromosome**						
Size (kbp)	5310	5264	5264	5444	5532	5440
%GC	51	51	51	50.7	50.7	50.7
No. CDSs	5194	5132	5131	5278	5735	5765
No. rRNA	22	22	22	22	22	22
No. tRNA	91	90	90	96	103	96
No. Prophages	14	12	12	17	18	17
No. Genomic islands (GI)	44	40	44	48	55	51
Accession number *	CP044313	CP044312	CP044311	CP044315	reference	reference
**Plasmid**						
Size (kbp)	175	70/45	99	74/53	95/52	68
%GC	47.6	48.7/48.8	47.2	49.1/49.2	47.2/48.8	49.2
Accession number *	CP051656	CP051654, CP051655	CP051653	CP051657, CP051658	reference	reference

***** The accession numbers are provided for the strains, including three environmental *E. coli* O45:H16 strains and one clinical *E. coli* O45:H2 strain (SJ7), which were sequenced in this study with the given accession numbers. Additionally, “reference” indicates the chromosome and plasmid of two clinical strains (FWSEC0003 and 2011C-4251) obtained from the National Center for Biotechnology Information (NCBI) with the accession numbers indicated in Appendix A. **^α^** The strains contained two different plasmids.

**Table 2 microorganisms-08-00782-t002:** The number of various virulence genes present in the chromosomes and plasmids of six STEC O45 strains analyzed in this study.

Virulence Genes	*E. coli* O45:H16			*E. coli* O45:H2		
	RM11911	RM13745	RM13752	SJ7	FWSEC0003	2011C-4251
**Chromosome**						
*stx* _1a_	1^α^	1	1	1	1	1
*stx* _2a_	-	-	-	-	-	1
*gad*	2	2	2	2	2	2
*iss*	2	2	2	1	2	1
*iha*	-	-	-	-	-	-
*iroN*	1	1	1	-	-	-
*lpfA*	1	1	1	-	-	-
*cif*	-	-	-	1	1	1
*eae*	-	-	-	1	1	1
*efa1*	-	-	-	1	1	1
*espA*	-	-	-	1	1	1
*espB*	-	-	-	1	1	1
*espF*	-	-	-	1	1	1
*espI*	-	-	-	1	1	1
*espJ*	-	-	-	-	-	-
*nleA*	-	-	-	1	1	1
*nleB*	-	-	-	3	3	3
*nleC*	-	-	-	1	1	-
*tir*	-	-	-	1	1	1
*tccP*	-	-	-	-	-	-
**Plasmid**						
*hyl*	1	1	1	1	1	1
*afa*	1	-	-	-	-	-
*cdt*	1	-	-	-	-	-
*esp*	1	1	-	-	-	-
*cnf1*	1	-	1	-	-	-
*iha*	1	1	-	-	-	-
*ehxA*	-	-	-	1	-	1
*etpD*	-	-	-	1	1	1
*stcE*	-	-	-	1	1	1
*toxB*	-	-	-	-	-	-
*katP*	-	-	-	-	-	-

**^α^** The numerical value indicates the number of genes present, and “-” means that no virulence gene was detected.

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
