# Peer review of "Is Shiga Toxin-Producing Escherichia coli O45 No Longer a Food Safety Threat? The Danger is Still Out There"

_microorganisms, 2020, doi:10.3390/microorganisms8050782_

Round 1
Reviewer 1 Report
The topic covered in the paper is very interesting. The methodology is described in detail and suitable to get the appropriate results and consistent with the conclusions.
The legends of figure 3 that are located at the top of the right corner are not legible.
In the paragraph where it approaches the results of phylogenetic analysis of stx prophages obtained, between lines 288 – 297, the style of writing can make these results difficult to follow by the readers. It would be recommendable to simplify or present it in a different order.
Author Response
We appreciate the reviewer’s comments.
Point 1: The legends of figure 3 that are located at the top of the right corner are not legible.
Response 1: We agree with the reviewer’s comments and have provided the updated figure 3 with a larger and clear legend on the right side of the figure in the revised manuscript.
Point 2: In the paragraph where it approaches the results of phylogenetic analysis of stx prophages obtained, between lines 288 – 297, the style of writing can make these results difficult to follow by the readers. It would be recommendable to simplify or present it in a different order.
Response 2: The appropriate changes have been made to simplify the sentences based on the reviewer’s comment in the revised manuscript (line 297-300).
Reviewer 2 Report
This is an excellent work and help food scientists to plan next experiments especially to approaches to reduce the risk of such pathogens.
few comments; please consider revising the second objective, there is something missing in the statement and it does not reflect the work. I also would like for you to include how this work fits into your long term research especially the use of natural antimicrobial work. The results from this will definitely direct our work to conduct future studies on the selection of novel antimicrobial compounds. Overall, this is great work. I gain good knowledge. Best of luck......SAI
Author Response
We appreciate the reviewer’s comments.
Point 1: Please consider revising the second objective, there is something missing in the statement and it does not reflect the work.
Response 1: The appropriate changes have been made based on the reviewer’s comment in the revised manuscript (line 82-83).
Point 2: I also would like for you to include how this work fits into your long term research especially the use of natural antimicrobial work.
Response 2: The additional information has been added based on the reviewer’s comment in the revised manuscript (line 403-409).